# Nrf2 in Neoplastic and Non-Neoplastic Liver Diseases

**DOI:** 10.3390/cancers12102932

**Published:** 2020-10-12

**Authors:** Claudia Orrù, Silvia Giordano, Amedeo Columbano

**Affiliations:** 1Department of Oncology, University of Torino, 10060 Candiolo, Italy; claudia.orru@ircc.it; 2Candiolo Cancer Institute-FPO, IRCCS, 10060 Candiolo, Italy; 3Unit of Oncology and Molecular Pathology, Department of Biomedical Sciences, University of Cagliari, 09042 Monserrato, Italy

**Keywords:** liver regeneration, acute and chronic liver injury, multistep hepatocarcinogenesis, Keap1/Nrf2 pathway

## Abstract

**Simple Summary:**

Although the Keap1-Nrf2 pathway represents a powerful cell defense mechanism against a variety of toxic insults, its role in acute or chronic liver damage and tumor development is not completely understood. This review addresses how Nrf2 is involved in liver pathophysiology and critically discusses the contrasting results emerging from the literature. The aim of the present report is to stimulate further investigation on the role of Nrf2 that could lead to define the best strategies to therapeutically target this pathway.

**Abstract:**

Activation of the Keap1/Nrf2 pathway, the most important cell defense signal, triggered to neutralize the harmful effects of electrophilic and oxidative stress, plays a crucial role in cell survival. Therefore, its ability to attenuate acute and chronic liver damage, where oxidative stress represents the key player, is not surprising. On the other hand, while Nrf2 promotes proliferation in cancer cells, its role in non-neoplastic hepatocytes is a matter of debate. Another topic of uncertainty concerns the nature of the mechanisms of Nrf2 activation in hepatocarcinogenesis. Indeed, it remains unclear what is the main mechanism behind the sustained activation of the Keap1/Nrf2 pathway in hepatocarcinogenesis. This raises doubts about the best strategies to therapeutically target this pathway. In this review, we will analyze and discuss our present knowledge concerning the role of Nrf2 in hepatic physiology and pathology, including hepatocellular carcinoma. In particular, we will critically examine and discuss some findings originating from animal models that raise questions that still need to be adequately answered.

## 1. Introduction

The Kelch-like ECH-associated protein 1 and nuclear factor (erythroid-derived-2)-like 2 (Keap1/Nrf2) pathway is one of the most important cellular defense systems against environmental stress. Nrf2 is a basic leucine zipper transcription factor belonging to the Cap N′ Collar (CNC) family that binds to DNA unique sequences, named antioxidant responsive elements (AREs) [1,2,3]. Keap1 is a substrate adaptor for a Cul3 dependent ubiquitin ligase complex responsible for the negative regulation of Nrf2 and other proteins [1]. Under normal or unstressed conditions Nrf2 is sequestered in the cytoplasm by Keap1, and rapidly degraded in the proteasome. The interaction between Keap1 and Nrf2 requires the Nrf2 DLG and ETGE domains: ETGE engages Nrf2 with Keap1, while the DLG motif acts as latch to (un)lock the Nrf2 Neh2 domain in an appropriate spatial orientation, and thus regulates Nrf2 stability and ubiquitination [4,5]. Hence, Keap1 mediates Nrf2 polyubiquitination and, therefore, its proteasomal degradation. Under oxidative stress, such as exposure to reactive oxygen species (ROS) or electrophiles, Keap1 undergoes conformational modifications, resulting in the loss of interaction between Nrf2 and Keap1 [4,5,6]. As a consequence, Nrf2 is not degraded, but moves to the nucleus where it promotes transcription of a number of genes involved in cellular protection, including phase I (such as NAD(P)H:quinone oxidoreductase 1 (NQO1) and heme oxygenase (HO)-1), phase II (glutathione *S*-transferase (GST)), and phase III enzymes [6,7,8,9]. By modulating a wide range of cell defense processes, Nrf2 enhances the cell’s capability to counteract harmful substances.

Considering that the liver is the major organ involved in chemical detoxification and drug metabolism, it is not surprising that Nrf2 is highly expressed in hepatocytes, to antagonize the high levels of ROS and, therefore, maintain liver homeostasis and favor cell survival [10]. In agreement, Nrf2 signaling attenuates the pathological features of chronic liver injuries by suppressing oxidative stress (due to the unbalance between increased ROS production and decreased antioxidant defenses), which is known to be the key player in the development and progression of chronic liver damage [11].

Nrf2 was originally considered as a pro-survival and tumor suppressor gene because of its cytoprotective function against exogenous and endogenous insults. However, several studies provided evidence that hyperactivation of the Nrf2 pathway creates an environment that favors the survival of DNA-damaged cells, protecting them against oxidative stress, chemotherapeutic agents, and radiotherapy, thus favoring cancer progression (Reviewed in [12,13]). The identification of a dark side of Nrf2 has generated new interest and some concern, because it is still unclear whether interference with Nrf2 results in a beneficial, or in a harmful, condition.

## 2. Nrf2 in Hepatic Regeneration

As the liver exerts an essential role in regulating the metabolic functions of the body, it must be able to rapidly and efficiently respond to, and cope with, metabolic perturbations. The liver’s capability to rapidly restore its mass has long been recognized as a crucial topic of study, both for experimental research as well as for clinical applications. Indeed, the efficient regenerative ability of the liver is required in conditions such as the transplantation of partial liver grafts, the living-related transplantation, or in those of elderly status displaying an impaired liver regenerative capacity [14,15,16]

Liver regeneration after 70% partial hepatectomy (PH) represents a classic and well-recognized experimental model of rapid, controlled, and reproducible cell proliferation in a mammalian organ system [17,18].

A relationship between liver regeneration and Nrf2 is now recognized, but very few studies have clarified the role and significance of this transcription factor in liver cell proliferation. Beyer et al. were the first to directly investigate the role of Nrf2 as a cell-cycle modulator during PH. Using *Nrf2*-lacking mice (KO), they observed a significant reduction in the number of proliferating hepatocytes at 48 h after PH, but a restauration of the liver mass similar to that of wild type (WT) animals at 5 days after surgery [19]. They concluded that *Nrf2* loss causes a delay of the regenerative process, possibly due to an impairment of IGF-R1 signaling. Puzzlingly, a delay in liver regeneration after PH was observed in the same group in mice bearing a constitutively active Nrf2 [20]. From a mechanistic perspective, they showed that the delay in regeneration was not due to the involvement of IGF-R1 signaling, as demonstrated in *Nrf2* KO mice, but rather to induction of p15 and Bcl2l1l. Contradictory results have also been obtained by Dai’s group, who showed that in *Keap1+/−* mice, who therefore overexpressed Nrf2, there was a delay of S phase entry [21]. In contrast to these finding, further work by the same group also showed that Nrf2 deficiency resulted in a delay of hepatocyte mitosis due to dysregulation of cyclin A2, Wee1, Cdc2, and cyclin B1 [22]. To add confusion, deregulation of cyclins also occurred in *Keap1+/−* [21]. It is worth mentioning that the contrasting results reported by Dai’s group were obtained in mice of the same strain and gender, and subjected to the same extent of surgery (70% PH). Overall, their results cast doubt on the real effect of Nrf2 in the regulation of cell cycle-related proteins. Irrespective of the mechanism, it remains totally obscure why either reduction or increase of Nrf2 leads to a similar delay of liver regeneration after PH.

An experimental study performed including WT, *Nrf2* KO, and Nrf2 constitutively activated mice (due to generation of either *Keap1* KO or expression of active Nrf2 mutants) could help clarify whether and how Nrf2 is involved in hepatic regeneration. Such an experimental setting would allow the analysis of the same pathway from different perspectives, and the identification of similarities and differences among the animal groups in the same experimental conditions (e.g., age and gender), avoiding any bias. It is worth mentioning that, in this context, attention should also be paid to the presence of intrahepatic shunt, as it has been shown that genetic manipulation could lead to the development of intrahepatic shunt in *Nrf2*-null mice [23], thus affecting the regenerative capacity of the liver [24].

Even less is known concerning liver regeneration post-surgery in rats. Previous work suggested that rat liver regeneration after PH occurs in a Nrf2-independent manner, as revealed by lower expression and reduced activity of Nrf2 targets, such as glucose-6-phosphate dehydrogenase (*G6pd*) [25]. On the other hand, a strong induction of the expression of Nrf2 target genes, such as the placental form of glutathione S-transferase (*Gstp*) and *Nqo1*, was observed when hepatocyte proliferation was stimulated by treatment with metals endowed with hepatomitogenic activity, such as lead nitrate (LN) [26]. This suggests that Nrf2 contribution to liver proliferation in rat liver could be dependent upon the nature of the proliferative stimulus; if so, it remains to be established whether Nrf2 is required for hepatocyte proliferation, or to handle the hepatotoxicity of exogenous mitogens.

Discrepancies can also be found when examining regeneration after carcinogen-induced acute liver injury. Ngo and collaborators found that *Nrf2*-null mice did not develop liver cancer after diethylnitrosamine (DEN) administration. They suggested that this effect was probably the consequence of an impairment of regeneration following DEN-induced liver necrosis [27]. In *Nrf2* KO rats, while the loss of *Nrf2* completely inhibited the early steps of DEN-induced hepatocarcinogenesis, liver regeneration post-liver injury induced by DEN was similar to that of *Nrf2* WT rats, suggesting that other mechanisms can sustain liver regeneration in the absence of Nrf2 [28]. Whether these differences are species-dependent remains elusive.

In conclusion, the role of Nrf2 in liver regeneration is far for being understood, and more studies are required.

## 3. Nrf2 in Acute Hepatotoxicity

The protective role of Nrf2 in hepatotoxicity has been well established. Indeed, *Nrf2*-KO mice were more susceptible to acetaminophen-induced acute liver injury than WT animals [29], and it has been proposed that this protective role is played by the p62/Nrf2/Keap1 axis [30]. Indeed, sulforaphane pre-treatment attenuated lipid peroxidation and liver failure after acetaminophen exposure by triggering Nrf2-target genes, further supporting the pivotal role of Nrf2 in the protection against hepatotoxicity [31]. In another experimental setting, hepato-specific deletion of Nrf2 sensitized cells to CCl_4_-induced liver damage and increased fibrosis and inflammation [32]. Accordingly, Klasseen and collaborators demonstrated that, in a condition of Nrf2 overexpression, cadmium- and/or furosemide-induced liver injury was attenuated compared to that observed in WT mice [33,34]. In hepatotoxicity induced by microcystins (toxins produced by certain freshwater blue-green algae) Nrf2 prevented glutathione depletion and protected against oxidative stress-induced liver damage [35]. In line with this report, Gu and co-workers reported that alpha lipoic acid attenuated microcystin-leucine arginine (M-LR)-induced hepatotoxicity through Nrf2 modulation and regeneration of glutathione synthesis [36].

Aflatoxin-B1 (AFB1) is one of the most potent genotoxic and hepatocarcinogenic agents. *Nrf2* KO rats are more sensitive to the toxic effects of AFB1 compared to WT animals [37]. Intriguingly, while the induction of cellular glutathione by the potent triterpenoid CDDO-Im (2-cyano-3, 12 dioxoolean-1,9-dien-28-oic acid-imidazole) attenuated AFB1-hepatotoxicity in WT rats, this effect was completely lost in *Nrf2* KO animals, suggesting that the beneficial effects of CDDO-Im are Nrf2-dependent [34]. This is in agreement with a study which identified Nrf2 relevance in the detoxification response after AFB1 exposure, suggesting a Nrf2 chemoprotective role [38]. In general, loss of Nrf2 has been demonstrated to increase the sensitivity to several other toxic chemicals, such as acetaminofen, carbon tetrachloride, 3,5-Diethoxycarbonyl-1,4-dihydrocollidine (DDC), and cadmium chloride, highlighting a critical role of this transcription factor and its downstream target genes in acute and chronic liver damage [39,40,41,42,43,44].

Altogether, these studies suggest that Nrf2 is a key factor in the protection against hepatic injuries induced by several toxic agents.

## 4. Nrf2 in Chronic Liver Injury

As mentioned before, oxidative stress has a pivotal role in the pathophysiology of several chronic liver diseases. Thus, Nrf2 involvement in the chronic liver damage occurring during viral hepatitis, and alcoholic and non-alcoholic fatty liver diseases is not surprising.

### 4.1. Nrf2 in Viral Hepatitis

Both HBV and HCV are known causes of hepatocellular carcinoma (HCC), through promotion of inflammation and oxidative stress in the liver [45]. Under these conditions, occurring liver damage is followed by fibrosis, cirrhosis, and HCC [46]. Among the mechanisms responsible for the ability of HBV to induce HCC, it has been proposed that the viral protein HBx interacts with p62 [47] which sequesters Keap1, preventing its interaction with Nrf2, and allowing Nrf2 stabilization and translocation to the nucleus [48]. Thus, the formation of the HBx-p62-Keap1 complex results in Nrf2 activation. As these results imply a possible role of Nrf2 in survival of viral hepatitis-infected cells, it would be interesting to explore whether Nrf2 modulation might be a therapeutic strategy in HBV infections.

As far as HCV is concerned, HCV-associated HCC has been reported to be linked to p62 accumulation, which promotes Nrf2-dependent metabolic reprogramming [49]. In line with these reports, Nrf2 knockdown markedly suppressed HCV infection in HCV-persistently-infected cell lines [50].

If these results suggest that Nrf2 activation favors survival of virus-infected cells and increases the risk of HCC, contrasting results stem from other studies showing that HCV impairs the regulation of antioxidant enzymes by affecting Nrf2-sMaf binding [51]. The latter data were interpreted by the authors to suggest that the lack of Nrf2 may pave the ground for the genomic alterations required for neoplastic transformation.

Consistently, the phytocompound Lucidone inhibited HCV activity by promoting Nrf2-mediated transcription of HO-1, which suppressed HCV replication through its product, biliverdin [52]. Similar results have been obtained by Tseng et al. who found that Celastrol, a quinone methide triterpene, triggered NS3/4A protease activity inhibition through HO-1 induction and, in turn, suppressed HCV replication [53]. In the same line, treatment with the antioxidant sulforaphane suppressed viral replication; this effect was mitigated by the knockdown of the Nrf2-target gene HO-1, implying the essentiality of HO-1 in sulforaphane treatment [54].

Taken together, there are contrasting reports about the implication of Nrf2 in viral hepatitis infection. On one hand, Nrf2 induction inhibits virus replication through its target genes (i.e., HO-1); on the other hand, HBV- and HCV-positive cells might exploit the protective function of Nrf2, and this could contribute to liver tumorigenesis [55].

### 4.2. Nrf2 in Alcoholic Fatty Liver Disease (AFLD)

Chronic alcohol consumption is a well-known risk factor for HCC development. An effect of prolonged ethanol uptake is the increase of ROS production that, together with cell death, increases proliferation and inflammation, contributing to the onset of liver cancer [56]. Several reports have shown the crucial role of Nrf2 in liver protection after alcohol exposure (reviewed in [57]). In ethanol-exposed mice, *Nrf2* genetic inactivation deeply worsened hepatic damage. Indeed, *Nrf2*-null mice displayed severe liver failure that led to death within 24 days upon an ethanol-supplemented regimen [58,59]. Besides the aggravated inflammatory response, *Nrf2*-null mice exhibited an accumulation of the toxic metabolite acetaldehyde. In a mirror experiment, hepatocyte-specific *Keap1* KO mice (endowed with higher Nrf2 activity) exhibited decreased serum alanine aminotransferase (ALT) levels after ethanol administration [60]. In support of the beneficial role of Nrf2 in alcohol-induced liver damage, pharmacological activation of this transcription factor attenuated alcohol-induced liver failure by decreasing ROS production and apoptosis [61,62].

Other works evidenced that Nrf2 accelerates alcohol-induced steatohepatitis. In fact, Wang and collaborators found that Nrf2-signalling positively modulated the expression of hepatic very low-density lipoproteins receptor (VLDLR), which contributes to the development of alcohol-induced liver injury [63]. In agreement, ethyl pyruvate attenuated AFLD by downregulating the Keap1/Nrf2 pathway [64].

It has been well established that cirrhosis is the end stage of alcoholic fatty liver disease, and further enhances the probability of HCC development [56]. Nrf2’s role in this stage of the disease is, at present, not clear. Ceni et al. reported that acetaldehyde treatment (used as a model of alcoholic liver fibrosis) triggered hepatic stellate cell (HSC) activation through the induction of oxidative stress [65]. By using the same model of HSC activation, Ni et al. found that treatment with IL-22 inhibited HSC proliferation in a Nrf2 dependent manner [66].

Even though these studies suggest an involvement of Nrf2 in AFLD progression, more comprehensive studies, with proper cellular and animal models, are needed to determine the specific role of Nrf2 in the different liver cell types.

### 4.3. Nrf2 in Non-Alcoholic Fatty Liver Disease (NAFLD)

Non-alcoholic fatty liver disease is rapidly becoming one of the most common causes of chronic liver disease worldwide, and it is a major cause of liver-related morbidity and mortality [67]. NAFLD begins with aberrant triglyceride accumulation in the liver and may progress to non-alcoholic steatohepatitis (NASH), cirrhosis, and HCC. Oxidative stress has been proposed as the major actor involved in NAFLD-NASH pathogenesis [6,68,69,70].

Studies by Yates et al. suggested that Nrf2 overactivation could play a role in lipid metabolism, leading to downregulation of genes involved in fatty acid synthesis [71]. The same group also demonstrated that treatment with CDDO-Im reduced fat accumulation by 40% in high-fat diet-(HFD)-fed mice, and that this effect was lost in *Nrf2* KO mice, supporting the notion that Nrf2 is involved in hepatic lipid homeostasis [72].

Notably, while WT mice “exclusively” exhibited steatosis when fed HFD, Nrf2 loss promoted the progression of steatosis to NASH, with severe liver injury and subsequent inflammation [73]. Similar results have also been obtained with another model of NASH, namely the methionine-choline deficient (MCD) diet, a regimen that induces fatty changes, liver damage, cirrhosis, and then HCC [74,75]. When fed an MCD diet, *Nrf2* null mice developed severe steatosis, showed enhanced ROS production, and worse inflammation and fibrosis compared to their WT counterparts, likely due to a decrease of antioxidant systems [76,77]. Further support for the protective effect of Nrf2 came from a recent study showing that Nrf2 overexpression suppressed the deleterious effect on NASH progression of the hepatocyte-specific *c-Met* KO (known to trigger NASH development in MCD-fed mice) [78,79].

Some works have shown that pharmacological activation of Nrf2 reverses steatosis and improves the pathological hallmarks of NASH. Indeed, Ezetimibe, a FDA-approved drug for the treatment of hypercholesterolemia [80], attenuated the pathological status of steatohepatitis through p62-dependent Nrf2 activation. Lee and colleagues showed that Ezetimibe activates the AMP-activated protein kinase (AMPK), which phosphorylates p62 at S351, thus allowing p62/Keap1 binding and subsequent Nrf2 activation and nuclear translocation [81]. Similarly, the administration of acetylenic tricyclic bis(cyano enone) (TBE-31) mitigated NASH by decreasing endoplasmic reticulum stress, oxidative stress, and inflammation in WT mice (but not in *Nrf2* null mice) previously treated with HFD, along with fructose-containing drinking water [82]. This underlines the requirement of Nrf2 in the mechanism responsible for the protective effect of TBE-31.

Therefore, targeting the Nrf2-pathway might be promising in the prevention and treatment of NAFLD. Nevertheless, the exact role of Nrf2 in several physiopathological processes of the liver still remains uncertain (Figure 1)

## 5. Nrf2 and Hepatocellular Carcinoma (HCC)

HCC is the most common primary malignancy of the liver [83] and the fourth most common cause of cancer-related death worldwide. HCC has a dismal prognosis, with the ratio between mortality and incidence being approximately 95% [84]. Unfortunately, at the moment, no effective systemic therapy exists for patients with advanced HCC, and the few FDA-approved multikinase inhibitors showed only an unsatisfactory improvement in survival [85]. For this reason, a more complete understanding of the molecular mechanisms involved in HCC development and progression is pivotal for improving therapeutic strategies for this lethal tumor. Liver cirrhosis represents one of the major risk factors for HCC development, and is often preceded by hepatitis of different etiologies, such as HCV, HBV, alcohol, and hypernutrition [86]. Since these conditions share the pivotal role of oxidative stress as the main pathogenic driver [87], a growing number of researchers have focused their attention on the role of the Keap1/Nrf2 pathway in HCC (for comprehensive reviews see [88,89]).

### 5.1. Nrf2 and Cancer Prevention

The possible anti-tumorigenic activity of Nrf2 in non-HCC models was originally proposed based on the cell-protective ability of this transcription factor. Indeed, mice knockout for the Nrf2 downstream target, *Nqo1*, displayed susceptibility to benzo(a)pyrene-induced tumor skin development [90]. Moreover, the role of Nrf2 in cancer prevention has been demonstrated in *Nrf2* null mice in which benzo(a)pyrene-induced gastric cancers occurred with higher frequency, compared to their WT counterpart [91]. In addition, while treatment of WT mice with Oltipraz (a synthetic dithiolethione, exerting its chemopreventive action through modulation of conjugating and detoxification enzymes such as GST, NQO1, and glutathione reductase [92,93]) reduced by 52% the number of tumors detected in the forestomach, its chemoprotective activity was completely lost in *Nrf2* null mice [91]. Similar results have been obtained in bladder cancer after the exposure of N-butyl-N-(4-hydroxybutyl)nitrosamine) (BBN) [94].

Modulation of the Nrf2 signaling deeply affects the initiation step of hepatocarcinogenesis, as confirmed by chemical-induced carcinogenesis experiments performed in *Nrf2* KO mice. Indeed, in these mice Kitamura and collaborators showed an increased sensitivity to 2-amino-3-methylimidazol (4,5-*f*)quinoline (IQ) hepatocarcinogenicity, possibly due to IQ accumulation as a consequence of decreased GST activity, followed by oxidative damage, and hepatic failure [95]. A higher susceptibility to DNA adducts, hepatotoxicity, and cancer development was also shown by other groups [13,96].

As a consequence of the role played by Nrf2 in antioxidative defense, Nrf2 activation should be pursued for preventing, not only chronic liver diseases, but also cancer development, especially at early stages of hepatocarcinogenesis [13].

In line with this hypothesis a series of in vitro studies suggested the chemopreventive activity of Nrf2, especially in the case of aflatoxin B1 (AFB1)-induced HCC. Sulforaphane, a phytochemical whose precursor is abundant in a variety of cruciferous vegetables, especially in broccoli sprouts, directly interacts with Keap1 cysteine residues resulting in Keap1 inactivation and activation of Nrf2 transcriptional program [97,98]. Accordingly, sulforaphane-mediated Nrf2 induction interferes with cell viability, as well as cell migration and colony formation in human hepatoma cells [99]. A series of clinical trials conducted in Qidong, China (a region where patients have high dietary exposure to AFB1), showed that Sulforaphane contained in broccoli sprouts increased aflatoxin detoxification, thus attenuating cancer risk [98].

Similar conclusions were also achieved in in vivo models of AFB1-induced liver cancer, using the Nrf2 activator Oltipraz. Kensler and co-workers investigated the dietary effect of Oltipraz in countering AFB1 metabolism, DNA adducts, and hepatic tumorigenesis in male F344 rats [100]. Oltipraz treatment resulted in a significant reduction in the liver area occupied by preneoplastic lesions. In the authors’ opinion, the protection against tumorigenesis is attributable to the inhibition of aflatoxin metabolism and to the enhancement of electrophile detoxification. Moreover, a long-term experiment performed by the same group demonstrated a complete protection of Oltipraz against AFB1-induced cancer associated with longer life span and improvement in survival [101]. The translational value of these observations was definitively assessed in a phase IIa trial performed in China, showing that high doses of Oltipraz strongly decreased the levels of Aflatoxin M1, the major phase 1 oxidative metabolite of aflatoxin [102]. CDDO-Im is a synthetic triterpenoid that induces a number of genes that are under control of Nrf2. Yates et al., who evaluated the chemopreventive activity of CDDO-Im, showed that it reduced the formation of hepatic preneoplastic lesions, and inhibited the hepatocarcinogenic process, reducing AFB1 adducts and increasing the mRNA levels of the detoxification enzymes [38]. The chemopreventive effect was completely lost in *Nrf2* KO mice, further suggesting that it is mediated by Nrf2-induced detoxification enzymes [38]. In the same line, Yamamoto’s group demonstrated that *Nrf2* loss strongly affected the susceptibility of rats to AFB1 toxicity, as shown by higher hepatotoxicity and DNA-adducts after a single exposure of AFB1, compared to their WT counterpart. Additionally, *Nrf2* KO rats were prone to lethality, as the CDDO-Im effect was completely lost, further confirming the notion that the CDDO-Im chemoprotective role is mediated by Nrf2 modulation [37].

Taken together, all these reports suggest that, in addition to its inhibitory effect on liver injury caused by hepatotoxins, activation of the Keap1/Nrf2 pathway might inhibit the first step of hepatocarcinogenesis (initiation step) by impairing the biotransformation of pro-carcinogens and, consequently, the formation of highly reactive metabolites that, through their binding to DNA, can induce mutations in genes relevant for cancer development (Figure 2).

### 5.2. The Other Face of Nrf2

Despite Nrf2 being originally described as a tumor suppressor, the question of whether Nrf2 activation is “good” or “bad” is still unanswered. Thus, the role of Nrf2 and its negative regulator, Keap1, has stimulated many studies in the field of cancer to find out whether the activation of this pathway could be a good strategy for cancer treatment.

The involvement of Nrf2 in liver cancer was already predictable from experimental studies performed during the 60s and 70s. One of the first pieces of evidence suggesting Nrf2 involvement in HCC emerged from reports in which the activity of D-T diaphorase (now known as Nqo1) and G6PD, both now well-known Nrf2 target genes, was increased in rat chemically-induced preneoplastic foci/nodules [103,104,105]. Later on, Nrf2 was shown to regulate the expression of GSTP, another long-known preneoplastic marker [106,107].

As time went on, a growing body of studies reported the “oncogenic” function of Nrf2 [108]. Besides the regulation of cytoprotective enzymes, in malignant cells Nrf2 was shown to promote metabolic reprogramming by inducing enzymes of the pentose phosphate pathway (PPP, such as G6PD) through the well-conserved AREs [109]. Nrf2 induces the non-oxidative phase of the PPP, through the modulation of the expression of transaldolase (TALDO) and transketolase (TKT), thereby directing carbon flux towards the PPP. This promotion of metabolic reprogramming occurs under sustained activation of the PI3K-AKT pathway, suggesting a positive feedback loop between these two signaling pathways [109]. Additionally, it was shown that in HCC, Nrf2 directly activates the transcription of methylenetetrahydrofolate dehydrogenase 1-like (MTHFD1L), leading to NADPH production [110]. Interestingly, most of the above observations made in in vitro experiments were confirmed in rat livers by Kowalik et al., who demonstrated that the induction of PPP by Nrf2 occurred from the early steps of hepatocarcinogenesis, and characterized the most aggressive subset of preneoplastic lesions, suggesting the role of Nrf2 as a key factor in liver cancer development [25].

Together with regulating metabolic pathways and proliferation, Nrf2 favors cell survival through the modulation of anti-apoptotic proteins. A research group in Baltimore hypothesized that Nrf2-mediated induction of anti-apoptotic proteins could play a role in the drug resistance of cancer cells. Indeed, Nrf2 promoted cell survival by inducing transcription of the anti-apoptotic proteins Bcl-2 [111] and Bcl-xL, and downregulation of the pro-apoptotic factors, Bax and caspase 3/7 [112]. In agreement with these studies, caspase-3 activity has been shown to significantly increase upon Nrf2 loss [113], further supporting the notion that Nrf2 has a role in controlling survival.

It is common knowledge that hepatocarcinogenesis is a multiphasic process, starting from preneoplastic foci to nodules, and progressing to early HCC and advanced HCC. This notion raised the relevant question of whether activation of the Keap1/Nrf2 pathway was an early event, possibly driving cancer development, or a change linked to the neoplastic transformation typical of late stages of cancer progression.

Unfortunately, a limiting factor in the study of late stages of hepatocarcinogenesis is that it is difficult to discriminate the molecular alterations that are the cause from those that are simply the consequence of cell transformation. Furthermore, the study of the early steps of human hepatocarcinogenesis is hampered by late diagnosis, mainly due to the lack of symptoms in the early phases [114]. Therefore, when and how Nrf2 interferes with the process of initiation, promotion, and progression of HCC is not fully understood. In this scenario, animal models are an essential tool to understand the involvement of Nrf2 in the pathogenesis of liver cancer.

Animal models allow the dissection of the several steps of hepatocarcinogenesis, such as the rat resistant-hepatocyte (RH) and the choline-methionine deficient (CMD) models [74,115], whose translational value have been established [116,117], have been very useful to address this question. By using these animal models, Columbano et al. showed that the dysregulation of the Keap1/Nrf2 pathway takes place in the earliest steps of hepatocarcinogenesis [28,117,118,119]. In addition, it was shown that the genetic inactivation of Nrf2 fully impaired the clonal expansion of carcinogen-initiated hepatocytes to preneoplastic nodules in rats [28], in spite of the presence of a tumor promoting environment, such as liver damage and compensatory hepatocyte proliferation caused by CMD diet. The inability of *Nrf2* KO to develop preneoplastic nodules supports the notion that Nrf2 is essential for the growth of carcinogen-altered hepatocytes [28]. In this line, studies with *Nrf2* KO mice showed that while DEN injection to WT newborn animals led to HCC development in 100% of the animals, no tumors were observed in *Nrf2* KO mice [27].

Intriguingly, however, while in both these studies *Nrf2* genetic inactivation led to inhibition of the tumorigenic process, there is discrepancy in the interpretation of such an inhibitory effect. Indeed, Ngo et al. suggested that the inhibition of the tumorigenic process was due to an insufficient compensatory proliferation following DEN-induced liver necrosis, and therefore, leading to an impairment of the initiation step [27]; on the other hand, no difference in DEN-induced DNA damage, DNA repair, liver necrosis, and compensatory regeneration was found in *Nrf2* KO and WT rats, suggesting that lack of Nrf2 did not affect initiation of hepatocarcinogenesis, but rather inhibited the promotion step by severely impairing the growth of initiated cells to a preneoplastic stage [28].

Irrespective of the mechanisms involved, from these studies it is evident that Nrf2 is mandatory for HCC initiation and progression. However, it is important to stress that, while Nrf2 seems to be essential for HCC development, it does not act as on oncogene when overexpressed in hepatocytes, as it is not sufficient to promote the transformation of liver cells [118,120]. In the same line, are the findings that *Keap1* knockdown mice display increased proliferation of forestomach epithelium, but not cancer [121], and that constitutive activation of Nrf2 (by deletion of its Neh2-Keap1-interaction domain) does not increase the rate of primary tumor formation in a mouse cancer model [122].

In human HCCs, support for a pro-tumorigenic role of Nrf2 stems from the observation that Nrf2 expression correlates with tumor size, poor differentiation, and presence of metastasis. In addition, high levels of Nrf2 are associated with decreased survival [123]. Moreover, high levels of activation of the Nrf2 pathway (evaluated as high NQO1 expression) in human HCC correlated with elevated α-fetoprotein, large tumor size, worse prognosis, and higher incidence of recurrence [124]. Similarly, a reduced expression of the Nrf2 inhibitor Keap1, was associated to poor 5-year overall survival and worse disease-free survival [125].

## 6. Mechanisms of Nrf2 Activation in HCC

Accumulating evidence showing constitutive Nrf2 activation in liver cancer prompted efforts to discover the mechanisms responsible for such dysregulation (Figure 3).

### 6.1. Somatic Mutations in Keap1 and Nrf2 Genes

A number of studies have reported that in human primary tumors *Nrf2* and/or *Keap1* mutations are present, located in the domains involved in the interaction between the two proteins; these mutations disrupt their interaction, leading to Nrf2 nuclear translocation and transcriptional activation [12,126].

Whole-exome sequencing revealed that *Nrf2* and *Keap1* mutations are present in 6–8% of human HCC, with differences in the prevalence of the mutated genes according to the different studies; interestingly, *Nrf2* and *Keap1* mutations are mutually exclusive [127,128,129].

Animal models allowed identification of *Nrf2* mutations as the earliest molecular changes responsible for the activation of the Keap1/Nrf2 pathway. Indeed, *Nrf2* mutations, but not *Keap1*, occurred at high frequency at the initial steps of rat hepatocarcinogenesis [28,118,119]. Notably, some of the *Nrf2* mutations found in rat hepatocarcinogenesis are those identified in human samples (e.g., D27, D29, E79, T80, E82) [130]. Intriguingly, the number of preneoplastic and neoplastic lesions carrying *Nrf2* mutations decreased along with the progression to malignancy [119]. The fact that mutations are one of the earliest events in hepatocarcinogenesis suggests that they may be relevant for the onset of HCC, as they likely confer a growth advantage to malignant cells.

Somatic mutations of *Keap1* appeared less frequently in the early stages and only in nodules non-mutated for *Nrf2* [118], supporting the notion that *Nrf2* and *Keap1* mutations are mutually exclusive.

### 6.2. The Crosstalk between Nrf2 and p62

Besides the canonical regulation, another Nrf2 activation mechanism depending on Keap1 was identified by two different groups in 2010 [48,131]. This “non-canonical” regulation is mediated by the p62 protein. P62, also known as sequestosome 1 and A170 (p62/SQSTM1/A170), is a selective autophagy adaptor. Autophagy is a multistep process characterized by the development of a phagophore, and its maturation into an autophagosome, which undergoes fusion with a lysosome for degradation and recycling of cellular components. Through its ubiquitin-interacting and LC3-interacting domains, p62 recognizes polyubiquinated proteins, and shuttles them for autophagic clearance [132]. P62 binds to the Keap1-Kelch domain through the Keap1-interacting-region (KIR), competing with Nrf2, and displacing the Keap1/Nrf2 interaction. Specifically, the phosphorylation of p62 on S351 increases p62 binding affinity for Keap1, leading to Keap1 sequestration and autophagy-mediated degradation [48,133]. As a result, Nrf2 is stabilized and translocated into the nucleus, where it induces the transcription of its target-genes, including p62, generating a positive feedback loop [134].

Mice models carrying autophagy defects proved a useful tool to understand the involvement of the Keap1-Nrf2-p62 axis in liver cancer. In a context of autophagy defect, the development of multiple tumors is driven by p62 accumulation and subsequent Keap1 sequestration. This is demonstrated by the fact that the genetic inactivation of p62 in mice fully impaired tumor formation, likely due to the restauration of the Keap1-Nrf2 binding and subsequent Nrf2 inactivation [135]. Accordingly, Nrf2 deletion prevented the formation of liver tumors, suggesting that Nrf2 is required for p62-driven tumor development [136,137].

Intriguingly, autophagy seems to play a dual role in HCC development, depending on the stage of tumorigenesis. Therefore, several studies have been performed to investigate the role of p62 in the different steps of hepatocarcinogenesis. A very recent study showed that in the CMD rat model of hepatocarcinogenesis, the p62-driven Nrf2 activation occurs at late stages [119]. This is in line with the literature reports that p62 accumulation is a late event in murine hepatocarcinogenesis [138]. Interestingly, in the CMD model no p62 could be observed in early preneoplastic foci/nodules carrying *Nrf2* mutations, while its accumulation occurred at late stages in HCC devoid of *Nrf2* mutation [119]. The progressive loss of mutations associated with a concomitant p62 accumulation implies that distinct mechanisms are responsible for Keap1/Nrf2 pathway activation at different steps of hepatocarcinogenesis (Figure 4). However, whether this is a general phenomenon, or is unique to the CMD model is unclear, as in the RH model, Kowalik et al. showed p62 accumulation in a subset of aggressive preneoplastic nodules, characterized by their positivity to the putative stem/progenitor cell marker cytokeratin-19 and a high frequency of *Nrf2* mutations [139]. In addition, Umemura et al. showed that p62 accumulation is a crucial pre-neoplastic event that leads to activation of Nrf2, and promotes liver cancer development in mouse models [140].

A p62-dependent Nrf2 regulation has also been demonstrated in human HCC, where p62 upregulation correlated to increased risk of HCC development [49,141,142]. Furthermore, Komatsu’s group demonstrated that p62 promoted HCV-associated liver tumorigenesis and chemoresistance through Nrf2-dependent metabolic reprogramming [49]. Hence, p62 targeting could be an important strategy in HCCs characterized by alteration of the Keap1-Nrf2-p62 axis. K67, a compound that interferes with the interaction between phospho-p62 and Keap1, resulting in the restoration of the Keap1-Nrf2 binding, has been shown to render cancer cells sensitive to anti-cancer drugs [49]. In line with this report, a K67 derivate has demonstrated its efficacy in inhibiting phospho-p62-Keap1 interaction in the Huh-7 HCC cell line, overcoming in vitro resistance to chemotherapeutic agents [142].

In the autophagy context, it is important to underline that, together with the canonical (ROS-dependent, KEAP1-dependent) and non-canonical (p62-dependent) mechanisms of NRF2 activation, endoplasmic reticulum (ER) stress represents an additional connection between autophagy and Nrf2 [143,144]. Indeed, eukaryotic translation initiation factor 2 alpha kinase 3 (PERK), a type I ER transmembrane protein which participates in the regulation of the Glucose regulatory protein 78 (GRP78) [145], in response to proteins accumulated in the ER lumen, is separated from Keap1 and phosphorylates Nrf2 [146], allowing its nuclear translocation, and activation of cell survival signals. Accordingly, persistent HCV replication in liver cirrhosis has been shown to activate NRF2 phosphorylation and nuclear translocation, suggesting that the PERK pathway could directly contribute to cell survival through NRF2 phosphorylation and to HCC development [147].

### 6.3. Nrf2 Regulation by MicroRNAs

MicroRNAs (miRNAs) are single stranded non-coding RNAs involved in the regulation of gene expression. MiRNAs post-transcriptionally inhibit mRNA translation into proteins, and can contribute to the regulation of diverse cellular processes [148,149]. In cancer, miRNAs have been proposed to promote or inhibit cancer progression as they can function as oncogenes or tumor suppressors [149].

Currently, there is growing interest in the potential role of miRNAs in modulating the Keap1/Nrf2 pathway. As far as HCC is concerned, it was shown that miR-141 conferred resistance against 5-Fluorouracil treatment in a Nrf2-dependent manner. Indeed, miR-141 interfered with Keap1 mRNA and protein stability, resulting in the constitutive activation and nuclear translocation of Nrf2 [150]. On the other hand, miR-144 reversed the acquisition of resistance to 5-Fluorouracil in the HCC cell line by targeting Nrf2 mRNA [151]. Shi and collaborators obtained similar results, demonstrating that miR-340 reversed cisplatin resistance of HCC cells by targeting Nrf2 [152]. In vivo studies also pointed to a role for miRNAs in the regulation of the Keap1/Nrf2 pathway. Indeed, rat preneoplastic nodules exhibiting Nrf2 upregulation also displayed upregulation of miR-200a, which directly targets Keap1 [117].

Overall, although the regulation of Nrf2 by miRNAs is a promising area of research, due to the very little information about the interaction between miRs and Nrf2 in HCC, the role of small non-coding RNAs in modulating the Keap1/Nrf2 pathway in liver tumors remains largely elusive.

### 6.4. Posttranslational Keap1/Nrf2 Modulation

It is common knowledge that metabolites, such as fumarate can interfere with Keap1-Nrf2 binding [153,154]. Fumarate is a Krebs cycle metabolite that acts as an electrophile, and reacts with Keap1 cysteine residues, causing its conformational modification [153,154]. The resulting succination of Keap1 leads to repression of Nrf2 ubiquitination and its nuclear translocation, thereafter. Similarly to fumarate, succinylacetone interferes with Nrf2-Keap1 binding by alkylating Keap1 Cys23, Cys319, Cys406, and Cys513, and resulting in Nrf2 stabilization [155].

Keap1 can also be inactivated by the tripartite motif-containing (TRIM)-25 protein that, in a scenario of endoplasmic reticulum (ER) stress, directly interacts with Keap1, promoting its ubiquitination and subsequent degradation. As a consequence, Nrf2 translocates to the nucleus and activates its transcriptional program. Thus, in the presence of ER stress TRIM25 promotes tumor cell survival and HCC growth through the modulation of Nrf2 signaling [156].

A novel mechanism by which Nrf2 sustains tumorigenesis is through SUMOylation. Indeed, defective Nrf2 SUMOylation inhibited HCC development [157]. Nrf2-SUMOylation is required for stimulation of de novo serine synthesis by Nrf2. In turn, serine enhances ROS clearance, and increases the level of Nrf2 SUMOylation, leading to sustained HCC growth.

A very recent and interesting study unveiled another possible mechanism of Nrf2 activation in HCC, showing that Nrf2 activity is dependent on Fructosamine-3-kinase (FN3K), a kinase responsible for protein de-glycation [129]. The authors showed that in the absence of FN3K, Nrf2 is glycated, becoming unstable and defective at binding to small MAF proteins. They also showed that HCC development depends on FN3K in vivo, and that *N*-acetyl cysteine treatment partially rescued the effects of FN3K loss. Overall, this study not only revealed an additional mechanism of Nrf2 activation, but also implicated FN3K as a potential target modulator of Nrf2 activity in cancer (Figure 3).

## 7. Nrf2 as a Mediator of Resistance

As mentioned before, HCC is one of the most aggressive tumors, and endowed with poor prognosis. To date, liver transplantation or surgical resection represents the main treatment options in patients at early stages of HCC. However, the diagnosis of HCC is too often made at late stages, when there is no effective treatment that can improve a patient’s outcome. A number of chemotherapeutic and cytotoxic agents have been used for treatment of liver cancer, such as Cisplatin, Docetaxel, and 5-Fluorouracil [158]. However, the cancer death rates keep rising due to the high resistance of HCC to conventional chemotherapies. The capability of malignant cells to resist both chemo- and radiotherapy is the main hurdle in the way of improvement in patient survival. The multidrug resistance (MDR) of cancer cells is a phenomenon characterized by the resistance to chemo-treatment with several anticancer drugs, resulting in the decrease of therapeutic efficacy. It has been reported that Nrf2 is frequently overexpressed in chemo-resistant HCC cells [158,159]. Zhou and colleagues demonstrated that Nrf2 expression increased when the BEL-7402 cell line was exposed to the 5-Fluorouracil, suggesting an involvement of this transcription factor in HCC chemoresistance [160]. The use of Camptotechin, a Nrf2 inhibitor, enhanced the sensitivity of HCC cell lines to chemotherapeutic agents, impairing cancer cell viability [161]. Furthermore, inhibition of Nrf2 through Apigenin reversed the drug-resistant phenotype. Mechanistically, Apigenin attenuates the resistance to Doxorubicin by reducing Nrf2 expression through downregulation of the PI3K/AKT pathway [162]. In addition, Nrf2 silencing suppressed chemoresistance to 5-Fluorouracil and Doxorubicin [163].

Sorafenib, the first FDA-approved molecular drug for the treatment of HCC, is an oral multikinase inhibitor that inhibits the Raf/MEK/ERK pathway and receptor tyrosine kinases [164]. The role of Nrf2 in sustaining resistance to sorafenib has been demonstrated in HCC cell lines [163]. Indeed, Nrf2 overexpression has been observed in Sorafenib-resistant cells, while its suppression of Nrf2 partially reverted resistance [163], possibly due to its role in mediating Sorafenib-induced ferroptosis [165].

Another mechanism through which Nrf2 could contribute to chemoresistance is the regulation of the multidrug resistance-associated proteins (MRPs) that enhance the efflux of chemotherapy drugs, thus decreasing drug accumulation within cells [158,159]. Experimental evidence supporting this hypothesis stems from the studies by Maher and collaborators, who observed that mouse liver Mrp2-6 are induced by Nrf2 through its binding to ARE sequences in the MRP promoter regions [166,167].

## 8. Nrf2 as a Molecular Target

Considering the described dual role of the Keap1/Nrf2 pathway in liver diseases, it is clear that either Nrf2 activation or inhibition could play a therapeutic role in the different pathologies.

### 8.1. Nrf2 Activators

As described, Nrf2 activation could provide a clinical advantage in conditions of acute/chronic liver toxicity during viral infections, and could play a chemopreventive role. To increase Nrf2 activity, several strategies and compounds have been envisaged [168].

The most successful Nrf2 activator to date is probably the fumaric acid, ester dimethyl fumarate (DMF), which is FDA-approved for psoriasis [169] and relapsing-remitting multiple sclerosis (MS) [170,171]. DMF reduced the number of peripheral CD8^+^ cells and B lymphocytes, leading to an anti-inflammatory shift in B cell responses [172], and was shown to activate Nrf2 in the central nervous system in a MS mouse model [173]. DMF-induced Nrf2 activation resulted in improved clinical course, axon preservation, and enhanced astrocyte activation. These DMF effects were Nrf2-dependent, as they were lost in Nrf2-null mice.

Oltipraz, an enhancer of GSH biosynthesis and phase II detoxification enzymes, is under phase III clinical trial for the treatment of nonalcoholic fatty liver disease (ClinicalTrials.gov Identifier: NCT04142749).

Ursodiol (ursodeoxycholic acid) is an FDA-approved drug for the treatment of primary biliary cirrhosis. Kawata and colleague showed enhanced hepatic Nrf2 activation after ursodeoxycholic acid treatment in patients with primary biliary cirrhosis [174].

Several electrophilic compounds (such as sulforaphane, curcumin, resveratrol, quercetin, genistein) are Nrf2 inducers as they modify Keap1 cysteins, leading to Keap1 conformational changes and prevention of Nrf2 degradation [175]. Their therapeutic value is currently under evaluation in different clinical settings).

### 8.2. Nrf2 Inhibitors

Even though Nrf2’s role in cancer is still debated, an increasing number of reports evidence the detrimental role of the anti-oxidative role of Nrf2 in liver carcinogenesis. This observation is in line with previous reports showing that administration of antioxidants per se, may accelerate the later stages of certain types of cancer [176,177], indicating that antioxidants have distinct effects on different stages of tumorigenesis.

Thus, Nrf2 inhibition has emerged as a promising therapeutic strategy, and many efforts have been made to identify inhibitors that could suppress the oncogenic role of this transcription factor (for exhaustive reviews on Nrf2 inhibitors see [178,179]).

Natural compounds derived from medicinal plants have long been proposed as therapeutic agents in a wide number of human diseases. Recently, many plant extracts have been suggested to play a role as anticancer agents. Not surprisingly, based on the notion that Nrf2 displays oncogenic activity, several efforts have been made to identify natural compounds with inhibitory properties on Nrf2 functions. Those listed below represent the best known and most used natural compounds to inhibit Nrf2 in the context of HCC: (see also Table 1).

(1) Brusatol. Ren and colleagues found that Brusatol, a quassinoid derived from *Brucea javanica*, enhances the efficacy of chemotherapy by depleting Nrf2 protein through a posttranscriptional mechanism [180,181]. These results suggested the effectiveness of using Brusatol to combat chemoresistance. However, although Brusatol was initially considered a specific Nrf2 inhibitor, two independent studies have shown that its effect is a consequence of the suppression of a wide range of proteins, and not of the specific inhibition of Nrf2 [182,183]. Thus, although Brusatol might be a useful tool, especially in cancer therapy, caution should be exercised against the possible side-effects of this drug.

(2) Flavonoids. Flavonoids are a class of natural compounds able to modulate Nrf2 signaling. Luteolin enhanced the sensitivity to cisplatin in cell-derived xenografts by decreasing Nrf2 expression [184]; this effect occurs in a Nrf2-dependent manner, since it is was completely lost in Nrf2-null mice. Similar results have been observed in experiments performed with wogonin, which downregulated Nrf2, enhancing sensitivity to chemotherapy in the HepG2 cell line [185].

Apigenin was shown to revert chemoresistance of HCC cell lines through downregulation of Nrf2 via the PI3K/AKT pathway [162]. Moreover, in vivo experiments demonstrated the synergistic effects of Apigenin and Doxorubicin, resulting in the inhibition of tumor growth and enhancement of cell death.

(3) Alkaloids. Tsuchida et al., identified Halofuginone, a quinazoline alkaloid, as a candidate Nrf2 inhibitor. Specifically, Halofuginone inhibits Nrf2 protein synthesis, via binding and inhibition of the prolyl-tRNA synthetase; and in combination with anticancer drugs, such as cisplatin or doxorubicin, enhances sensitivity to anti-cancer therapy and reduces chemo- and radio-resistance [186]. However, Halofuginone, similar to Brusatol, is not a specific Nrf2 inhibitor as it represses global protein synthesis, thus rapidly depleting Nrf2. Camptotechin, another alkaloid isolated from the Chinese tree Camptotheca acuminate, inhibits Nrf2 activity in HCC cells, reverting chemoresistance [161]. However, its mechanism of action is not clear, as no direct effect in promoting Nrf2 protein degradation could be provided.

(4) All trans-retinoic acids (ATRA). Contrary to other Nrf2 inhibitors, retinoic acid does not impair Nrf2 expression, but it interferes with the Nrf2-ARE binding. Retinoic receptor alpha (RXRα) can form a complex with Nrf2, hindering Nrf2-ARE binding [187]. However, since retinoic acids elicit Nrf2 modulation in a concentration-dependent manner, further studies should be performed before considering retinoic acids as potential clinical candidates. It should also be noted that, although several studies reported that ATRA inhibits HCC development, no reports have demonstrated that this effect was mediated through Nrf2 inhibition.

(5) ARE Expression Modulator 1 (AEM1). Bollong and collaborators showed that in A549 cells, AEM1 decreases Nrf2-target gene expression without interfering with Nrf2 or Keap1 protein levels [188]. However, experiments performed in different cell lines revealed that the action of AEM1 is restricted to cells harboring constitutive Nrf2 activation, likely due to Nrf2 or Keap1 mutations [179]. Thus, its potential therapeutic use could be restricted to a subset of patients.

(6) ML385. Biswal’s group showed that ML385, a thiazole-indoline compound, can directly interact with the Neh1 domain of Nrf2, thus hampering the binding between Nrf2 and DNA [179,189]. Similarly to AEM1, ML385 exhibited specificity and selectivity in cell lines displaying Keap1 mutations.

(7) PHA767491. PHA767491 is considered a potent Nrf2 inhibitor, hindering its nuclear translocation, and resulting in tumor growth inhibition. However, PHA767491 is also a potent inhibitor of cyclin-dependent kinase (CDK) [190]. Thus, it is unclear whether the effect on tumor growth is due to Nrf2 or to CDK inhibition, or to the combined inhibitory effect on Nrf2 and CDK.

### 8.3. Additional Strategies to Inhibit the Keap1/Nrf2 Pathway

An additional strategy to inhibit Nrf2 signaling may consist in enhancing the negative control exerted by Keap1, either by increased synthesis or by promoting its physical interaction. In this regard, it is interesting to note that K67, a small noncovalent inhibitor of phospho-p62/Keap1 interaction, which thus increases Keap1-induced Nrf2 degradation, showed an antineoplastic effect in HCC cell lines by decreasing proliferation and enhancing cytotoxicity of anti-cancer drugs. Based on these findings, this inhibitor might be useful to treat HCC displaying Nrf2 hyperactivation consequent on p62 accumulation [49].

Altogether, these reports, although promising, suggest that the path to identifying an effective Nrf2 inhibitor is still long. Even though these compounds could hinder Nrf2 signaling, their clinical usefulness is hampered by the possible opposite, context-dependent role of Nrf2. Moreover, most of them are not Nrf2-specific.

## 9. Concluding Remarks

Growing interest in Nrf2 research accumulates each year, together with the awareness that our understanding of the physiopathological role of this transcription factor is far from complete. A recent review article has identified Nrf2 as a hallmark of cancer, by virtue of its many cellular/tissue functions [108]. Some of these functions have been thoroughly investigated (i.e., sustained proliferative signaling, protection from acute/chronic liver injury, both tumor suppressive and tumor promoting activity, and induction of metabolic reprogramming). Nevertheless, as our knowledge increases, the dark sides emerge more and more frequently, and several gaps in our understanding must still be filled.

Within the frame of the liver, in particular, the role of Nrf2 is still obscure in some of the functions mentioned above. A few examples are listed below:

(1) Sustained proliferation signaling: in spite of the many studies showing a stimulatory effect of Nrf2 on cell proliferation, works aimed to directly investigate the role of Nrf2 in liver regeneration proved unsatisfactory, as either the excess (Nrf2 overexpressing mice) or the lack of Nrf2 (*Nrf2* KO mice) caused a delay in liver regeneration. Same inconclusive results have been obtained using *Keap1* knockdown when opposed to Nrf2 deficient animals. The reasons for these apparently puzzling results are totally unclear.

(2) Protection from acute/chronic liver injury: While several studies have shown that Nrf2 induction exerts an inhibitory effect of liver injury induced by chemicals, recent work in mice and rats has shown the extent of DEN-induced liver cell necrosis in KO vs. WT animals. This is unexpected as metabolic biotransformation of DEN generates electrophiles that should cause more liver damage in the absence of the Nrf2-dependent antioxidative system. This finding raises the question of whether induction of Nrf2 must necessarily precede the chemical insult in order to be able to protect the hepatocytes from death or initiation of hepatocarcinogenesis.

As to the role of Nrf2 in chronic liver damage, while its protecting effect is evident in NAFLD, research on whether its induction limits or promotes the necro-inflammatory environment generated by HCV hepatitis gave rise to contradictory results.

(3) Nrf2 as a tumor suppressor or an oncogene: It is the general opinion that Nrf2, originally described as a tumor suppressor, also functions as an oncogene. Nrf2 pathway hyperactivation can help transformed/malignant cells to escape oxidative stress by promoting the expression of antioxidant target genes, sustain chemoresistance, prevent the intracellular accumulation of drugs in cancer cells, and subsequently protect the cells from apoptosis; on the other hand, an increased Nrf2 expression can be beneficial in conditions when chemoprotection is required (Figure 5).

Despite the inconsistencies and unanswered questions described above, owing to its critical role in non-neoplastic and neoplastic liver diseases, Nrf2 is an attractive target. However, the choice between Nrf2 inhibitors or activators should be made carefully and dependent on liver conditions. Nrf2 activators could be used for preventing acute liver damage, or ameliorating chronic hepatic injury in well characterized conditions, whereas Nrf2 inhibitors could be used for cancer treatment. Unfortunately, to date, there is only one FDA-approved Nrf2 activator (dimethyl fumarate), and no safe and specific Nrf2 inhibitors.

## Figures and Tables

**Figure 1 cancers-12-02932-f001:**
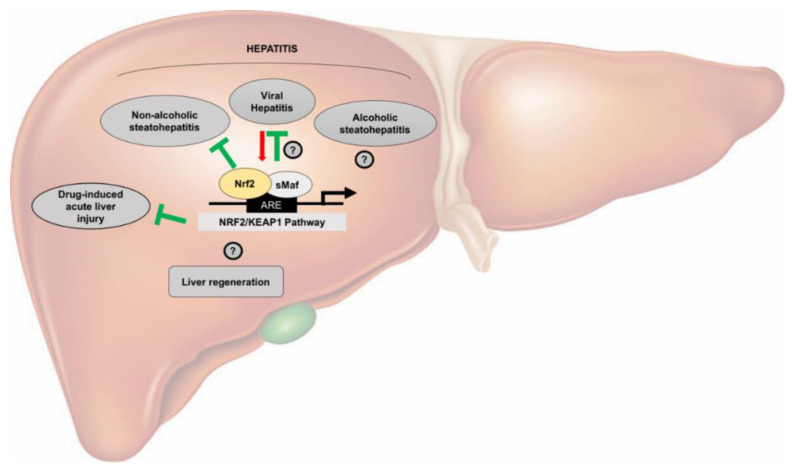
Nrf2 activation in non-neoplastic liver physiopathology. This scheme illustrates the role of the Keap1/Nrf2 pathway in non-neoplastic liver. For details see text. Green lines indicate inhibition; the red arrow indicates stimulation. Contradictory results are indicated by question marks.

**Figure 2 cancers-12-02932-f002:**
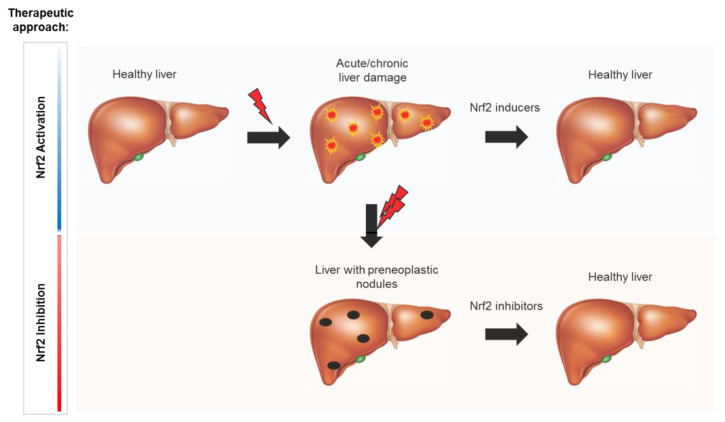
Schematic representation illustrating the opposite effects of Nrf2 activation on normal and carcinogen-initiated hepatocytes. Nrf2 inducers could lead to a more efficient repair of liver injury following administration of most necrogenic agents, including chemical carcinogens. On the other hand, Nrf2 inhibitors could promote reactive oxygen species (ROS)-induced injury, leading to death of preneoplastic cells, thus interfering with their progression to hepatocellular carcinoma (HCC). 
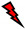
 stress promoting stimuli.

**Figure 3 cancers-12-02932-f003:**
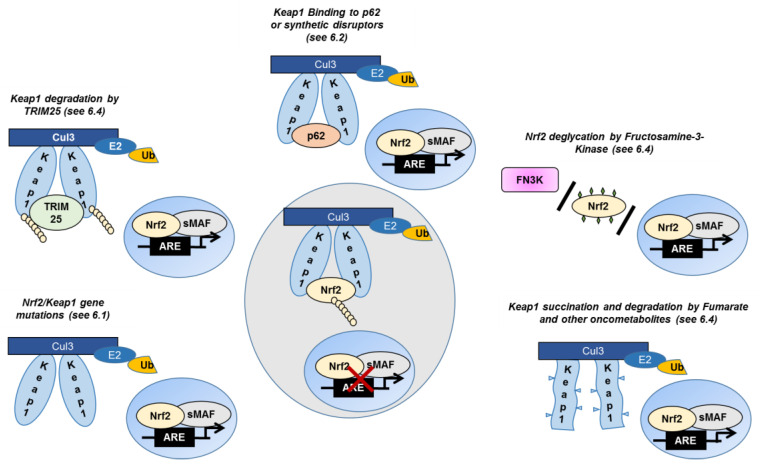
Distinct mechanisms leading to Nrf2 activation in cancer cells. The figure illustrates some of the best known mechanisms leading to the activation of the Keap1/Nrf2 pathway. Details are illustrated in the text. In the center of the figure is depicted an unstressed condition, where Nrf2 is bound to Keap1 and targeted to proteasomal degradation; nuclear translocation cannot thus take place. Numbers in brackets indicate the paragraph where the mechanism is described.

**Figure 4 cancers-12-02932-f004:**
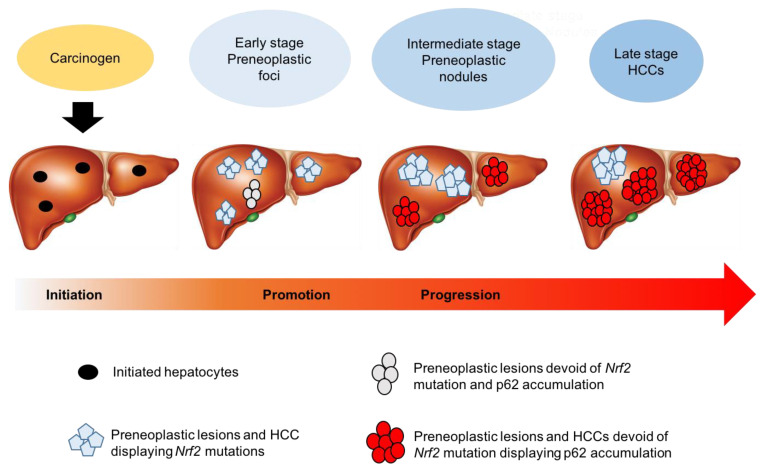
Schematic representation illustrating the possibility that distinct mechanisms of Nrf2 mutation occur at different stages of hepatocarcinogenesis. While in rat models [121] a high frequency of *Nrf2* gene mutations occurred in early preneoplastic lesions, their incidence diminished with the progression to malignancy, concomitantly with p62 accumulation, suggesting that activation of the Keap1/Nrf2 pathway at late stages is mainly due to Keap1 sequestration by p62.

**Figure 5 cancers-12-02932-f005:**
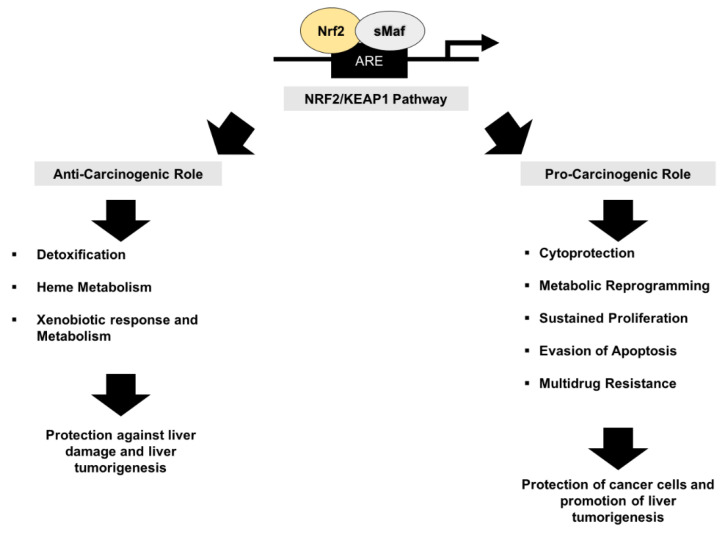
Dual role of Nrf2 in cancer.

**Table 1 cancers-12-02932-t001:** Compounds targeting the Keap1/Nrf2 pathway.

Compound	Molecular Target	Mechanism of Action	Effect	Experimental Model	Use	References
**Sulforaphane**	Keap1	Interferes with Keap1-Nrf2 binding	Nrf2 activation	*in vitro*	Liver Cancer	[99]
**DMF**	Keap1	Interferes with Keap1-Nrf2 binding	Nrf2 activation	*in vitro/in vivo*	Multiple Sclerosis	[173]
**Oltipraz**	Keap1	Interferes with Keap1-Nrf2 binding	Nrf2 activation	*in vivo*	Gastric Cancer	[91]
					Bladder Cancer	[94]
					Liver Cancer	[100,101,102]
**CDDO-Im**	Keap1	Interferes with Keap1-Nrf2 binding	Nrf2 activation	*in vivo*	Liver Cancer	[37,38]
**Curcumin**	Keap1 p38	Interferes with Keap1-Nrf2 binding	Nrf2 activation	*in vitro*	Renal epithelial cells	[175]
**Ursodiol**	Not specified	Not specified	Nrf2 activation	Human biopsies	Primary biliary cirrhosis	[174]
**Apigenin**	PI3K/Akt	Decreases Nrf2 mRNA and protein levels	Nrf2 inhibition	*in vitro/in vivo*	Liver Cancer	[162]
**Camptotechin**	Nrf2	Decreases NRF2 mRNA and protein levels	Nrf2 inhibition	*in vitro*	Liver Cancer	[161]
**Brusatol**	Overall protein translation	Decreases Nrf2 protein levels	Nrf2 inhibition	*in vitro/in vivo*	Lung Cancer	[180]
**Halofuginone**	Overall protein translation	Decreases Nrf2 protein levels	Nrf2 inhibition	*in vitro/in vivo*	Lung Cancer Esophageal squamous Carcinoma	[186]
**Wogonin**	Nrf2	Decreases Nrf2 protein levels Increases Keap1 protein levels	Nrf2 inhibition	*in vitro*	Liver Cancer	[185]
**Luteolin**	Nrf2	Decreases Nrf2 transcriptional activity	Nrf2 inhibition	*in vivo*	Lung Cancer	[184]
**ATRA**	Neh7 domain of Nrf2	Decreases Nrf2 transcriptional activity	Nrf2 inhibition	*in vitro*	Breast Cancer	[187]
**ML385**	Neh1 domain of NRF2	Decreases Nrf2 transcriptional activity	Nrf2 inhibition	*in vitro/in vivo*	Lung Cancer	[189]
**AEM1**	Nrf2	Decreases Nrf2 transcriptional activity	Nrf2 inhibition	*in vitro*	Lung Cancer	[188]
**PHA767491**	Nrf2	Decreases Nrf2 transcriptional activity	Nrf2 inhibition	*in vitro*	Liver Cancer	[190]
**K67**	Keap1	Interferes with Keap1-P62 binding	Nrf2 inhibition	*in vitro*	Liver Cancer	[49]

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
