# Peer review of "Nrf2 in Neoplastic and Non-Neoplastic Liver Diseases"

_cancers, 2020, doi:10.3390/cancers12102932_

Round 1

Reviewer 1 Report

This review article focused on the protective and adverse effects of Keap1/Nrf2 pathway in liver physiology. Keap1/Nrf2 pathway plays a central role in oxidative stress response and drug metabolism in organisms. The authors provides an overview of protective role of Keap1/Nrf2 pathway in several liver diseases, such as hepatitis and drug-inducible acute liver injury. The authors also discussed the light and dark sides of Keap1/Nrf2 pathway during hepatocellular carcinoma development.
The paper is well-described about Keap1/Nrf2 pathway function in liver under normal and pathophysiological conditions. The reviewer thinks the topic of the paper would be interesting for readers, recommends for publication in Cancers after minor revisions.

All minor comments are incorporated into the manuscript file attached.

Author Response

Reviewer 1.

The paper is well-described about Keap1/Nrf2 pathway function in liver under normal and pathophysiological conditions. The reviewer thinks the topic of the paper would be interesting for readers, recommends for publication in Cancers after minor revisions.

We thank the reviewer for her/his positive comments. All the criticisms have been addressed. Changes are highlighted in yellow. pP62 has also been substituted with p62 in figure 3.

Reviewer 2 Report

This is a very-well written review on the role of Nrf-2 in liver diseases. The images are informative and very well-designed. I think overall this is an interesting paper to read and it would be a good addition to the Molecular Biology section of the journal Cancers. I have provided a few comments below, which I consider minor revisions that could easily be fixed in a short time-span.

General remarks:

There has been a fairly recent review paper on the role of Nrf2 in HCC, which was also published in Cancers in 2018. The authors should specify in their response letter how their review moves the field forward, compared to the already existing review papers on this topic.

Try to include to most recent literature on Nrf2 in the review. This is a fast-moving field in HCC-research. The following papers on Nrf2 in liver cancer and/or liver cancer cell lines have been published in 2020, and they should therefore be discussed in the review: PMID: 32927432; PMID: 32839090; PMID: 32830144; PMID: 32780006; PMID: 32774486; PMID: 32751896; PMID: 32751080; PMID: 32694997; PMID: 32273945

The authors explain the link between autophagy and Nrf2, however they do not mention endoplasmic reticulum stress, which is an important aspect of both autophagy and Nrf2 signaling. There is also of evidence on the crosstalk between Nrf2-signaling, ER-stress and autophagy (reviewed here: PMID: 23800989 and PMID: 31351428). This would be specifically relevant to mention in the context of HCC, as ER-stress is an important contributor to liver cirrhosis and HCC (PMID: 27226027; PMID: 30610118; PMID: 24222102).

Specific remarks:

The first paragraph on liver regeneration (line 62-64) is out of place in the current form, as it only mentions liver regeneration as an animal model. If the aim of the authors is to give an introductory paragraph on liver regenerations, the authors should expand this section with some general information on liver regeneration in physiological and pathological conditions. This would increase the readability and avoid the occurrence of a short “floating” paragraph in the text.

The authors make a great statement about the dual role of NRF2 in liver regeneration and clearly show that both reduction or increase of Nrf2 can lead to a delay of liver regeneration after PH. It would improve the review, if the authors could also provide a short explanation (or hypothesis) why this is the case. What where the differences between the studies? Was the same % of PH used? Same strain of animals? These differences are hinted at in the text between line 82 and 89, as the authors elegantly propose alternative studies to solve the discrepancies. But I think it would benefit if some differences between the studies are already highlighted before.

The first two paragraphs on HBV and HCV (line 142 - 149) are out of place in the current form. Similar as the issue with liver regeneration, the authors provide a random fact about HBV and HCV, without sufficient context to provide a good introduction on the role of HBV and HCV in liver disease. It is indeed true that HBV and HCV are among the greatest risk factors for HCC-development. However, it is also one of the greatest contributors to liver cirrhosis. Therefore, the first paragraph should be rephrased to encompass a general statement on HBV and HCV in chronic liver disease, cirrhosis and HCC. Then secondly, I don´t see how the three mechanisms that are responsible for HBV-induced HCC-development are relevant for the content of the manuscript. The mechanisms of HBV-induced HCC are complicated and beyond the scope of the manuscript. A suggestion is to write that one the mechanisms responsible for the ability of HBV to induce HCC is through the presence of the viral protein HBx and expand a bit more on that, since this seems to be more relevant for the Nrf2 story.

The text on Nrf2 and HCC does not add anything relevant to the review.  Line 250-267 just explains the molecular basis of HCC, without a clear link to Nrf2.  Line 268 and 272 just emphasizes the role of Nrf2 in cirrhosis, which is the major risk factor of HCC and has already been discussed in the other subsections. Therefore, I suggest that this entire section should be removed from the review. If the authors find the information on the molecular basis of HCC useful, they could mention the specifics in the relevant parts about the crosstalk between Nrf2 and different genes.

Line 391-398: can the difference between the studies be explained through the different mouse models that were used? The authors should comment on this, especially since they are referring to their own study.

Line 406 – 412: the authors could also include information from the human protein atlas concerning expression of Keap1/Nrf2 expression in HCC. At a first glance, it seams like expression of Keap1 is prognostic in liver cancer

The text about the molecular targets would benefit from a table, thereby clearly mentioning the disease in which the drug has been tested (as not all therapies that are mentioned in the text are limited to liver disease), the mechanism of action (activator, inhibitor, specific target) and whether it was in vitro, in vivo or clinical (and in the latter case, the clinical trial number should be mentioned in the table). This will make it a lot easier to visualize the current evidence.

I do not see the added value of adding “high-throughput screening” as a separate subsection. Whether a compound was identified through screening of compound libraries or by specifically testing one compound is irrelevant for this review. The authors should incorporate this information in the other parts of the text.

Small remarks

Please proofread the manuscript carefully, there are still quite a few typos and/or grammatical errors. I included a list below, but there are for sure more issues with the spelling that need to be fixed.

Line 11: space between and and Silvia

Line 20: I would rephrase as “It remains unclear what the main mechanism behind the sustained activation of … “ to increase readability and flow in the abstract.

Line 24: It is not necessary to abbreviate HCC in the abstract, as this acronym is not further used in this part.

Line 30: Write the complete pathways name before the acronym in parentheses.

Line 33: binds to unique DNA sequences

Line 59: remove ´ before dark

Line 74: Space between BcL2 and reference. Period between the reference and contradictory.

Line 100: DENA lacks the full-term. Most people use the acronym “DEN”, although DENA is not incorrect.

Line 166: rephrase “in researchers´opinion” as it is unclear who the authors are refering to (themselves or the researchers in reference (51))

Line 230: Ezetimibe: consistently use capitals or non capitals (eg. Line 228)

Line 274 – 284: The authors should state in the first sentence that this evidence if from non-HCC models.

Line 360: rephrase:  “Together with regulating metabolic pathways”

Line 382: Columbano et al

Line 382: in the earliest steps

Line 412: this sentence is grammatically incorrect and needs to be reformulated.

Line 678: The same

Author Response

Reviewer 2.

This is a very-well written review on the role of Nrf-2 in liver diseases. The images are informative and very well-designed. I think overall this is an interesting paper to read and it would be a good addition to the Molecular Biology section of the journal Cancers. I have provided a few comments below, which I consider minor revisions that could easily be fixed in a short time-span

We thank the reviewer for her/his very encouraging comment.  

 There has been a fairly recent review paper on the role of Nrf2 in HCC, which was also published in Cancers in 2018. The authors should specify in their response letter how their review moves the field forward, compared to the already existing review papers on this topic.

We are aware that other reviews on Nrf2 and HCCs have been recently published. However, beside the excellent review by the Taniguchi’s group, most of them discussed the role of the Keap1-Nrf2 pathway in cancer cell lines which do not quite recapitulate the hepatocarcinogenic process. In our review, we wished to emphasize the significance of Nrf2 activation in different steps of the multistage hepatocarcinogenic process, as alterations at early stages of tumorigenesis are most likely those driving the process of cancer development. In addition, unlike other recent reviews, we discussed the role of alterations of the Keap1-Nrf2 pathway in the proliferation of normal liver cells (still largely unclear) and in non-neoplastic diseases, highlighting the many unanswered and often contradictory results present in the literature.

 Try to include to most recent literature on Nrf2 in the review. This is a fast-moving field in HCC-research. The following papers on Nrf2 in liver cancer and/or liver cancer cell lines have been published in 2020, and they should therefore be discussed in the review: PMID: 32927432; PMID: 32839090; PMID: 32830144; PMID: 32780006; PMID: 32774486; PMID: 32751896; PMID: 32751080; PMID: 32694997; PMID: 32273945.

According to the reviewer, we have included two recently published reviews (Int J Mol Sci 2020; Cancers 2018).

 The authors explain the link between autophagy and Nrf2, however they do not mention endoplasmic reticulum stress, which is an important aspect of both autophagy and Nrf2 signaling. There is also of evidence on the crosstalk between Nrf2-signaling, ER-stress and autophagy (reviewed here: PMID: 23800989 and PMID: 31351428). This would be specifically relevant to mention in the context of HCC, as ER-stress is an important contributor to liver cirrhosis and HCC (PMID: 27226027; PMID: 30610118; PMID: 24222102).

As suggested by the reviewer, we have inserted in the revised manuscript a statement concerning the link between ER stress, autophagy and Keap1-Nrf2 activation (lines 481-491)

 Specific remarks:

The first paragraph on liver regeneration (line 62-64) is out of place in the current form, as it only mentions liver regeneration as an animal model. If the aim of the authors is to give an introductory paragraph on liver regenerations, the authors should expand this section with some general information on liver regeneration in physiological and pathological conditions. This would increase the readability and avoid the occurrence of a short “floating” paragraph in the text.

According to the reviewer suggestion, we have expanded the paragraph concerning the role of Nrf2 in liver regeneration (lines 64-69).

 The authors make a great statement about the dual role of NRF2 in liver regeneration and clearly show that both reduction or increase of Nrf2 can lead to a delay of liver regeneration after PH. It would improve the review, if the authors could also provide a short explanation (or hypothesis) why this is the case. What where the differences between the studies? Was the same % of PH used? Same strain of animals? These differences are hinted at in the text between line 82 and 89, as the authors elegantly propose alternative studies to solve the discrepancies. But I think it would benefit if some differences between the studies are already highlighted before.

The reviewer is posing a very intriguing question. Indeed, there is no doubt that the results regarding the role of Nrf2 in liver regeneration are at least contradictory. Following the reviewer comment, we checked the experimental conditions associated to the different experiments performed by the groups of Dai (Ref 21,22) and Werner (Ref 19,20). As far as the Dai’s group is concerned, there is no difference in strain, gender, age of the animals, or the type of surgical procedure used in the two publications; on the other hand, the paper by Werner’s group does not provide any indication on strain, gender or age of the TGs mice used in the Hepatology paper (Ref 20). Therefore, it is difficult to provide an explanation for the opposite results obtained by the latter group. In conclusion, it appears almost impossible to give a scientific explanation to the contradictory results of these experiments. This is why we believe that a comprehensive experiment under strictly controlled experimental conditions should be done in order to establish the role of Nrf2 in liver regeneration.

The first two paragraphs on HBV and HCV (line 142 - 149) are out of place in the current form. Similar as the issue with liver regeneration, the authors provide a random fact about HBV and HCV, without sufficient context to provide a good introduction on the role of HBV and HCV in liver disease. It is indeed true that HBV and HCV are among the greatest risk factors for HCC-development. However, it is also one of the greatest contributors to liver cirrhosis. Therefore, the first paragraph should be rephrased to encompass a general statement on HBV and HCV in chronic liver disease, cirrhosis and HCC. Then secondly, I don´t see how the three mechanisms that are responsible for HBV-induced HCC-development are relevant for the content of the manuscript. The mechanisms of HBV-induced HCC are complicated and beyond the scope of the manuscript. A suggestion is to write that one the mechanisms responsible for the ability of HBV to induce HCC is through the presence of the viral protein HBx and expand a bit more on that, since this seems to be more relevant for the Nrf2 story.

The reviewer is correct. Accordingly, we have cut this section discussing only the link between HBx and Nrf2.

The text on Nrf2 and HCC does not add anything relevant to the review.  Line 250-267 just explains the molecular basis of HCC, without a clear link to Nrf2. Line 268 and 272 just emphasizes the role of Nrf2 in cirrhosis, which is the major risk factor of HCC and has already been discussed in the other subsections. Therefore, I suggest that this entire section should be removed from the review. If the authors find the information on the molecular basis of HCC useful, they could mention the specifics in the relevant parts about the crosstalk between Nrf2 and different genes.

 According to the reviewer comment, we removed the entire section and introduced a few sentences to better highlight the link between Nrf2 and HCC and to introduce some of the reviews suggested by the reviewer.

Line 391-398: can the difference between the studies be explained through the different mouse models that were used? The authors should comment on this, especially since they are referring to their own study.

The experiments cited in lines 391 to 398 were performed in different species: rats (Orrù et al, ref. 28) and mice (Ngo et al, Ref 27). This is specified in the text.

Line 406 – 412: the authors could also include information from the human protein atlas concerning expression of Keap1/Nrf2 expression in HCC. At a first glance, it seems like expression of Keap1 is prognostic in liver cancer.

As requested, we have investigated the human protein atlas to verify the prognostic value of Keap1 and NRF2 expression. As shown in the graphics below, expression of either Keap1 or NRF2 does not correlate with prognosis in liver cancer. This is somehow expected as their regulation takes place at post-translational rather than transcriptional level. However, a negative correlation can be found for the NRF2 target genes G6pd and Nqo1, meaning that activation of this pathway associates with a negative prognosis. These graphs are provided for the reviewer only (see attachment). We have not included them in the text as the results are in line with what has already been published, namely, the fact that NRF2 activation in HCC has a negative impact. However, we do not have problems in introducing them, if requested.

The text about the molecular targets would benefit from a table, thereby clearly mentioning the disease in which the drug has been tested (as not all therapies that are mentioned in the text are limited to liver disease), the mechanism of action (activator, inhibitor, specific target) and whether it was in vitro, in vivo or clinical (and in the latter case, the clinical trial number should be mentioned in the table). This will make it a lot easier to visualize the current evidence.

According to the reviewer’s suggestion, a table has been added (Page 16).

I do not see the added value of adding “high-throughput screening” as a separate subsection. Whether a compound was identified through screening of compound libraries or by specifically testing one compound is irrelevant for this review. The authors should incorporate this information in the other parts of the text.

We modified the heading as suggested by the reviewer.

Small remarks

Please proofread the manuscript carefully, there are still quite a few typos and/or grammatical errors. I included a list below, but there are for sure more issues with the spelling that need to be fixed.

We carefully rechecked the manuscript. We hope that the text now fulfils all the requisites

Line 11: space between and and Silvia

Done

Line 20: I would rephrase as “It remains unclear what the main mechanism behind the sustained activation of … “ to increase readability and flow in the abstract.

Done

Line 24: It is not necessary to abbreviate HCC in the abstract, as this acronym is not further used in this part.

Done

Line 30: Write the complete pathways name before the acronym in parentheses.

Done

Line 33: binds to unique DNA sequences

Done

Line 59: remove ´ before dark

Done

Line 74: Space between BcL2 and reference. Period between the reference and contradictory.

Done

Line 100: DENA lacks the full-term. Most people use the acronym “DEN”, although DENA is not incorrect.

Done

Line 166: rephrase “in researchers´opinion” as it is unclear who the authors are refering to (themselves or the researchers in reference (51)

Done

Line 230: Ezetimibe: consistently use capitals or non capitals (eg. Line 228)

Done

Line 274 – 284: The authors should state in the first sentence that this evidence if from non-HCC models.

Done

Line 360: rephrase: Together with regulating metabolic pathways”

Done

Line 382: Columbano et al

Done

Line 382: in the earliest steps

Done

Line 412: this sentence is grammatically incorrect and needs to be reformulated.

Done

Line 678: The same

Done

Reviewer 3 Report

In this review article, the authors describe the dual role of the Keap1/Nrf2 pathway in liver diseases from a wide range of aspects. This comprehensive approach nicely demonstrates the current landscape of Nrf2 research focusing on liver diseases, further indicate an important question: whether Nrf2 activation is ‘good’ or ‘bad’ regarding its therapeutic effects. I recommend the study for publication after the following revisions.

Major comments

On Page 9, line 403, the authors describe that Keap1 knockout mice display increased proliferation of forestomach epithelium indicating reference [123]. Although the indicated study investigated Keap1 knockdown mouse (Keap1flox/-mouse), they did not demonstrate the data of Keap1 knockout mice. Total knockout of Keap1 in mice leads to malnutrition and death at weaning as a result of esophageal hyperkeratosis mediated with maximal activation of Nrf2 by Keap1 knockout; it is important to distinguish between Keap1 knockdownmouse and Keap1 knockoutmouse. Hence, it should be revised to Keap1 knockdown mouse. In the same way, Keap1 KO (Page 16, line 679) also needs to be revised.

Minor comments

Some modifications in Figure 2 and Figure 3 would be necessary to help readers’ understanding.

Figure 2

  • The authors need to indicate what ‘Activation’ and‘Inhibition’ represent in the figure. Instead of ‘Activation’ and ‘Inhibition’, ‘Nrf2Activation’ and ‘Nrf2 Inhibition’ is better.
  • Although the lightning symbols shown in the figure are assumed to be presenting some kinds of stress mediated with the liver damage, it is not clear. The authors need to give a simple explanation for them.

Figure 3

  • It is suggested that the cellular localization of Keap1-Cul3 complex and Nrf2 translocated into the nucleus might be indicated in each diagram of Figure 3. The illustration of cytoplasmic membrane could be helpful to visually show it.
  • I think the author could indicate the corresponding numbers of paragraphs in each diagram of Figure 3 (g.6.1Nrf2/Keap1 gene mutations, 6.2Keap1 Binding to P62 or synthetic disruptors, 6.4Keap1 degradation by TRIM25, Nrf2 deglycation by Fructosamine-3-Kinase, Keap1 succination and degradeation by Fumarate and other oncometabolites).

Minor editing is required:

Page 4, line 162. Font size needs to be changed.

Page 6, line 252, 260, 262, Page 15, line 631. There are missing letters.

Page 15, line 614. ‘was’ needs to be deleted.

Author Response

Reviewer 3.

In this review article, the authors describe the dual role of the Keap1/Nrf2 pathway in liver diseases from a wide range of aspects. This comprehensive approach nicely demonstrates the current landscape of Nrf2 research focusing on liver diseases, further indicate an important question: whether Nrf2 activation is ‘good’ or ‘bad’ regarding its therapeutic effects. I recommend the study for publication after the following revisions.

We thank the reviewer for her/his very positive comments

Major comments

On Page 9, line 403, the authors describe that Keap1 knockout mice display increased proliferation of forestomach epithelium indicating reference [123]. Although the indicated study investigated Keap1 knockdown mouse (Keap1flox/-mouse), they did not demonstrate the data of Keap1 knockout mice. Total knockout of Keap1 in mice leads to malnutrition and death at weaning as a result of esophageal hyperkeratosis mediated with maximal activation of Nrf2 by Keap1 knockout; it is important to distinguish between Keap1 knockdown mouse and Keap1 knockout mouse. Hence, it should be revised to Keap1 knockdown mouse. In the same way, Keap1 KO (Page 16, line 679) also needs to be revised.

The reviewer is correct. We are sorry for not having properly quoted the papers where Keap1 knockdown were used. This mistake has been corrected in the revised manuscript.

Minor comments

Some modifications in Figure 2 and Figure 3 would be necessary to help readers’ understanding.

Figure 2

  • The authors need to indicate what ‘Activation’ and ‘Inhibition’ represent in the figure. Instead of ‘Activation’ and ‘Inhibition’, ‘Nrf2Activation’ and ‘Nrf2 Inhibition’ is better.
  • Although the lightning symbols shown in the figure are assumed to be presenting some kinds of stress mediated with the liver damage, it is not clear. The authors need to give a simple explanation for them.

Figure 3

  • It is suggested that the cellular localization of Keap1-Cul3 complex and Nrf2 translocated into the nucleus might be indicated in each diagram of Figure 3. The illustration of cytoplasmic membrane could be helpful to visually show it.
  • I think the author could indicate the corresponding numbers of paragraphs in each diagram of Figure 3 (6.1Nrf2/Keap1 gene mutations, 6.2Keap1 Binding to P62 or synthetic disruptors, 6.4Keap1 degradation by TRIM25, Nrf2 deglycation by Fructosamine-3-Kinase, Keap1 succination and degradeation by Fumarate and other oncometabolites).

All the changes suggested by the reviewer have been incorporated in the revised manuscript

Minor editing is required:

Page 4, line 162. Font size needs to be changed.

Done

Page 6, line 252, 260, 262, Page 15, line 631. There are missing letters.

Done

Page 15, line 614. ‘was’ needs to be deleted.

Done
